# Biosynthesis of Ribose-5-Phosphate—Metabolic Regulator of *Escherichia coli* Viability

**DOI:** 10.3390/cells14221775

**Published:** 2025-11-12

**Authors:** Tatyana A. Seregina, Rustem S. Shakulov, Irina Yu. Petrushanko, Konstantin V. Lobanov, Alexander S. Mironov

**Affiliations:** Engelhardt Institute of Molecular Biology, Russian Academy of Science, 119991 Moscow, Russia; isinfo@eimb.ru (R.S.S.); irina-pva@mail.ru (I.Y.P.); alexmir_98@yahoo.com (A.S.M.)

**Keywords:** pentose phosphate pathway, ribose-5-phosphate, anabolic sensor, genetic modifications of PPP, cell viability

## Abstract

Biosynthesis of ribose-5-phosphate (R5P) underlies all biosynthetic processes associated with biomass growth. Actively dividing cells continuously require building blocks for genome replication, synthesis of ribosomes and other derivatives containing R5P as a carbohydrate backbone. The main source of R5P in the cell is the pentose phosphate pathway (PPP), which is an anabolic sensor designed to coordinate the level of pentose phosphates and reduced NADPH required for anabolic processes. This review is devoted to a comparative analysis of R5P biosynthesis pathways among different domains of microorganisms, the features of PPP regulation in bacterial cells depending on physiological conditions, as well as genetic modifications of PPP and their effect on cell viability. We emphasize that ribose metabolism is a factor in the consolidation of cellular homeostasis under conditions of intensive biomass growth and the discrepancy between the processes of ribose synthesis and consumption is marked by spontaneous cell death.

## 1. Introduction

One of the fundamental components of cellular metabolism, along with glycolysis (the Embden–Meyerhof–Parnas pathway) and the tricarboxylic acid cycle (the Krebs cycle), is the pentose phosphate pathway (PPP). Since Otto Warburg discovered Zwischenferment (now glucose-6-phosphate dehydrogenase (Zwf, G6PDH)) and its cofactor NADP^+^, an enormous amount of research has been conducted [1], resulting in the canonical PPP map that we can find in the pages of modern biochemistry textbooks. Traditionally, two branches are distinguished in the PPP: irreversible oxidative, in which the reactions of ribulose-5-phosphate (Ru5P) synthesis are accompanied by the restoration of NADPH and the loss of CO_2_, and reversible non-oxidative is source of the exclusive metabolites ribose-5-phosphate (R5P), erythrose-4-phosphate (E4P) and sedoheptulose-7-phosphate (S7P)—precursors for the synthesis of nucleotides, aromatic amino acids and the cell wall component ADP-heptose in bacteria, respectively (Figure 1).

The mainstream of modern research has become the study of the role of PPP enzymes in various pathological processes, such as oncogenesis, infections, neurodegenerative diseases [2,3,4]. Modification of PPP for microbiological production of metabolites such as D-ribose and aromatic amino acids, which used in the creation numerous pharmaceuticals and bioactive compounds, including antitumor drugs, also represents a large area of scientific interest [5,6]. PPP is primarily considered as a system for protecting cells from oxidative stress, being the main point of generating NADPH to maintain the pool of reduced glutathione [7,8,9,10]. Oxidative stress can be induced by factors of various nature, including the action of antibiotics [11,12]. In our recent studies, we found a sharp decrease the tolerance to antibiotics for *E. coli* cells with a modified PPP, in which the processes of NADPH restoration and R5P biosynthesis are separated [13,14]. At the same time, there was no direct correlation between the level of oxidative stress and cell sensitivity to antibiotics. The main factor directly associated with an increase in the bactericidal effect of antibiotics was the R5P level.

This review is devoted to a comparative analysis of the features of R5P biosynthesis among different domains of microorganisms. The main regulatory mechanisms coordinating the work of the canonical PPP on the example of bacteria are considered in the context of two alternative pathways for the synthesis of pentose phosphates, catabolic (oxidative branch) and anabolic (reversed non-oxidative branch) (Figure 1). Using the example of various genetic modifications of PPP, we highlight an unexpected physiological feature of the anabolic synthesis of R5P as a stress resistance factor.

## 2. Diversity of R5P Biosynthetic Pathways

According to modern concepts, life on Earth arose through chemical evolution, which resulted in the emergence of complex molecules with the properties of storing and replicating information in the form of RNA and DNA polymers [15,16]. It is assumed that under prebiotic conditions, nucleotide synthesis occurred through abiotic reactions [17]. Ribose as a component of the sugar-phosphate backbone of nucleic acids can be formed in the widely known formose reaction or Butlerow reaction, during which formaldehyde self-condenses in the presence of alkali metal hydroxides to form a mixture of aldose and ketose sugars [18]. From a multitude of possible reactions, those that led to the emergence of catalytic networks capable of reproduction with the subsequent formation of a stable living system were selected during evolution [19]. The synthesis of ribose and other pentose phosphates in living organisms is one of the most ancient metabolic pathways and has a pre-enzymatic origin, which is confirmed by studies of the reconstructed reaction environment of the prebiotic Archean ocean [20]. Currently, this cascade of biochemical reactions catalyzed by complex enzymes is called the non-oxidative branch of PPP (Figure 2a).

The non-oxidative branch of PPP is present in all living organisms from archaea to higher eukaryotes and retains a central metabolic role in providing the cell with the precursor of nucleic acids R5P. The reactions of the non-oxidative branch of PPP overlap with the Calvin cycle in photosynthetic organisms, as well as with the assimilation pathways of C1-carbon compounds in methylorthophores, where pentoses act as carbon acceptors (Figure 2b,c) [21,22]. The sequence of reactions in the Calvin–Benson cycle—from triose phosphates to pentose phosphates—is opposite to the usual direction of non-oxidative PPP. The photosynthetic (anabolic) direction is supported by the replacement of the transaldolase step of the non-oxidative branch by the aldolase reaction with the formation of the intermediate sedoheptulose-1,7-bisphosphate (S1,7bisP) (Figure 2b) [21]. This anabolic direction of the non-oxidative branch occurs in several situations besides photosynthesis. Typical examples are representatives of the human intestinal microflora *Prevotella*, *Bacteroides*, *Firmicutes*, *Proteobacteria*, *Verrucomicrobia* and *Lentisphaerae*, which are able to metabolize C5 carbohydrates as the sole carbon sources [23]. Remarkably, these microorganisms do not have transaldolase, one of the main enzymes of the canonical non-oxidative branch of PPP. It has been shown that the synthesis and/or metabolization of pentose phosphates in intestinal microorganisms existing under anaerobic conditions is carried out via the S1,7bisP pathway (SBPP) involving pyrophosphate-dependent phosphofructokinase and fructose-bisphosphate aldolase (Figure 2d) [23,24].

**Figure 2 cells-14-01775-f002:**
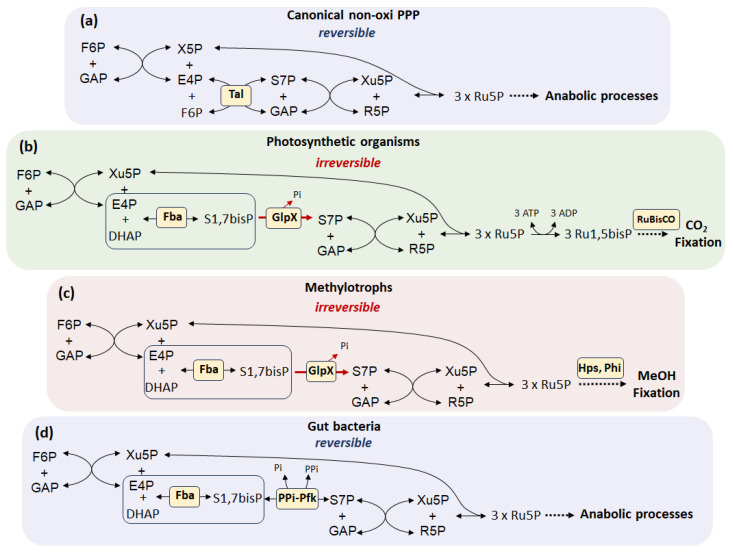
The non-oxidative branch of PPP in different groups of living organisms [23]. (**a**) The canonical branch of the PPP is completely reversible due to the transaldolase reaction and allows regulation of the intracellular level of R5P. (**b**,**c**) In organisms that fix C1 molecules, the direction of the non-oxidative branch toward the synthesis of pentose phosphates, which act as acceptors, is maintained by the activity of GlpX phosphatase, which dephosphorylates S1,7bisP. The reverse reaction of S7P phosphorylation requires ATP. (**d**) A feature of the non-oxidative branch of gut bacteria is the replacement of the transaldolase reaction by the aldolase activity with the formation of S1,7bisP from E4P and DHAP. The reversibility of this mode of the non-oxidative branch is provided by PPi-dependent phosphofructokinase. The formation of S1,7BP is of fundamental importance for the reversibility of the non-oxidative branch (indicated in a «box»). Abbreviations: F6P, fructose-6-phosphate; E4P, erythrose-4-phosphate; DHAP, dihydroxyacetone phosphate; GAP, glyceraldehyde-3-phosphate; Ru5P, ribulose-5-phosphate; Ru1,5P, ribulose-1,5-bisphosphate; R5P, ribose-5-phosphate; X5P, xylulose-5-phosphate; S7P, sedoheptulose-7-phosphate; S1,7bisP, sedoheptulose-1,7-bisphosphate.

It should be noted that an important function of both conventional PPP and SBPP is to allow the conversion of pentoses in both directions.

Many archaea also lack the classical oxidative and non-oxidative branches of PPP. Sequencing of archaeal genomes has revealed a virtual absence of genes encoding orthologs of the enzymes glucose-6-phosphate dehydrogenase (Zwf), 6-phosphogluconate dehydrogenase (Gnd), and 6-phosphogluconate lactonase (Pgl) [25]. R5P synthesis in methanococci occurs exclusively via enzymes of the non-oxidative branch acting in the “reverse” direction (from F6P and GAP to Ru5P), and not by oxidative decarboxylation of G6P. In addition, members of the archaeal domain that do not possess all the enzymes of the non-oxidative branch use an alternative ribulose monophosphate (RuMP) pathway to form pentoses required for nucleic acid biosynthesis, converting fructose-6-phosphate to Ru5P followed by formaldehyde assimilation (Figure 2c) [26,27].

Until recently, the RuMP pathway was thought to be present exclusively in methylotrophic bacteria for formaldehyde fixation. However, orthologs of *hps* and *phi* were recently cloned from *Bacillus subtilis* and shown to encode the enzymes 3-hexulose-6-phosphate synthase Hps and 6-phospho-3-hexuloisomerase Phi (Figure 2c) [28]. Further sequence analysis reveals that orthologs of *hps* and *phi* are widely distributed among bacterial and archaeal genomes [29].

Thus, the synthesis of pentose phosphates apparently arose as a cascade of reversible reactions of the non-oxidative branch of PPP, while the alternative pathway with the formation of the metabolite S1,7bisP with the irreversibility of the flow towards pentoses acquired a special anabolic significance in the processes of fixation of CO_2_ and other C1 hydrocarbons.

The emergence of photosynthetic microorganisms led to an increase in oxygen concentration and the accumulation of photosynthetically fixed carbon in the form of a storage polymer—glycogen [30]. During the study of metabolic processes underlying the dark/light transition in cyanobacteria, it was shown that the main pathway for glycogen breakdown in the dark is the oxidative branch of PPP, promoting the activation of key enzymes of the Calvin cycle, modulating the level of NADPH [31]. Apparently, the emergence of the oxidative branch of PPP was associated with the appearance of molecular oxygen as a by-product of photosynthesis and the need to protect the enzymatic apparatus of the cell from oxidative damage. Thus, there was a coupling of the generation of reducing equivalents of NADPH and the synthesis of pentose phosphates. In contrast to the reactions of the non-oxidative branch of PPP, which can be carried out non-enzymatically, the key reactions of the oxidative branch, concerning the interconversion of glucose-6-phosphate to 6-phosphogluconate, were not observed in Archean ocean simulations [20], which indirectly indicates an evolutionarily later emergence of the oxidative branch of PPP as a source of pentose synthesis.

## 3. Regulation of PPP

The canonical PPP is essentially an anabolic sensor designed to coordinate the level of pentose phosphates required for the maintenance of the cellular biosynthetic apparatus and genome replication with the generation of reducing equivalents of NADPH involved in the synthesis of amino acids, lipids and other metabolites during cell growth and division, depending on environmental conditions and nutrient availability. This position in central metabolism implies complex dynamic regulation through hierarchical interactions between the transcriptome, proteome and metabolome.

### 3.1. Regulation of the Oxidative Branch

The oxidative branch of PPP is evolutionarily newer than the non-oxidative one, but its activity is quite high in many pro- and eukaryotic organisms. The formation of the oxidative branch is apparently associated with the need to couple the regulation of redox balance in the cell with its anabolic requests. The transformation of the glycolytic/gluconeogenic metabolite glucose-6-phosphate into Ru5P occurs with the participation of two key enzymes, glucose-6-phosphate dehydrogenase (Zwf, Zwischenferment) and 6-phosphogluconate dehydrogenase (Gnd), which require NADP^+^ cofactors (Figure 1) [32]. The intermediate step of 6-phosphogluconolactone conversion to 6-phosphogluconate occurs both spontaneously and with the participation of 6-phosphogluconolactonases Pgl. However, it has recently been shown that intracellular non-enzymatic hydrolysis of 6-phosphogluconolactone does not compensate for the absence of Pgl [33]. It is generally accepted that the main positive transcriptional regulator of the *zwf* gene is the SoxS factor, belongs to the two-component SoxRS system, which is a fast and sensitive defense mechanism under oxidative stress, where the SoxR protein is a sensor of superoxide anions (O_2_^−^) [34]. It should be noted that the SoxRS system is quite conserved and is widespread among various groups of microorganisms, including pathogens [35]. Thus, the *zwf* gene, encoding the first enzyme of the oxidative branch of PPP Zwf, is part of the stress *soxRS* regulon, which allows the cell to generate the required amount of NADPH reducing agents with the ability to strictly control the level of other PPP metabolites [36].

On the other hand, the *zwf* gene is positively regulated by the transcription factors MarA and Rob [37,38]. The regulators MarA and Rob play a key role in the mechanisms of multidrug resistance in bacteria by activating the expression of genes encoding nonspecific transporters such as AcrAB-TolC, biofilm formation, synthesis and remodulation of cell wall LPS, thereby reducing the effectiveness of a wide range of antibiotics (tetracycline, chloramphenicol, ampicillin, nalidixic acid and rifampin) [39]. The transcription factors MarA, SoxS and Rob are involved in coordinating the response to stresses of various natures and are considered as a single regulon, the work of which is coordinated through transcriptional interaction [40]. Transcription of the *gnd* gene encoding 6-phosphogluconate dehydrogenase, which carries out the reaction of 6-phosphogluconate decarboxylation (Figure 1), is positively regulated by the GatE factor, which is involved in the response to stress caused by acids [41,42]. Thus, the oxidative branch of PPP is often considered a key player in the systems of rapid response to stress, and its catabolic function associated with the synthesis and maintenance of the pentose phosphate pool acquires secondary meaning.

Negative factors regulating the expression of genes of oxidative branch enzymes are the catabolite repressor/activator Cra and the oxygen-sensitive transcription factor FNR, which switches from aerobic to anaerobic growth of bacteria [43]. Under conditions of glucose limitation and a decrease in the fructose-1-phosphate pool, Cra activates the gene expression of gluconeogenic enzymes phosphoenolpyruvate synthetase Pps and fructose-1,6-bisphosphatase Fbp, Krebs cycle enzyme isocitrate dehydrogenase Icd and the glyoxylate shunt enzymes isocitrate lyase AceA and malate synthase A AceB, and simultaneously inactivates the expression of *zwf* and genes of the Entner–Doudoroff pathway enzymes phosphogluconate dehydratase Edd and aldolase Eda, redirecting the carbon flow towards gluconeogenesis and, as a consequence, the synthesis of R5P is carried out through the anabolic pathway from fructose-6-phosphate and glyceraldehyde-3-phosphate by reversing the non-oxidative branch of PPP [44]. Under anaerobic growth conditions, a similar situation is apparently observed with the participation of the FNR factor, which inactivates the expression of not only the *zwf* gene, but also *gnd* and induces the synthesis of pentoses in an anabolic manner [43]. Thus, during growth on gluconeogenic carbon sources, as well as under anaerobic conditions, the main pathway for the synthesis of pentose phosphates becomes the non-oxidative branch acting in an anabolic direction, where the key role belongs to the transketolase (Tkt) [45].

The coordination of metabolic fluxes between glycolysis and the oxidative branch of PPP has enormous physiological importance. It should be noted, that the availability of the precursor for both pathways, glucose-6-phosphate, and the Michaelis–Menten constants for the enzymes Zwf (K_m_ = 174 µM) [46] and glucose-6-phosphate isomerase (Pgi) (K_m_ = 1018 µM) are not the only factors determining the direction of flow [47]. All intermediates of PPP have been shown previously to inhibit the activity of the glycolytic enzyme Pgi. This was shown for 6PG, X5P, R5P, Ru5P, E4P and for S7P [48,49,50,51]. It is evident that there is a negative feedback mechanism through PPP at the Pgi/Zwf branch point between glycolysis and PPP itself [52]. On the other hand, the NADP^+^/NADPH ratio and the ATP level are the factors determining the intensity of the flux through the oxidative branch at the metabolome level. The level of oxidized NADP^+^ depends on the intensity of anabolic processes and the redox status of the cell. Thus, there is a competition between the enzymes Zwf (K_m_ for NADP^+^ 20 µM) and Gnd (K_m_ for NADP^+^ 49 µM) for the NADP^+^ cofactor and, therefore, the formation of Ru5P in the oxidative branch depends on the availability of oxidized NADP^+^. 6-phosphogluconate is another switching point for the flux through the oxidative branch, where in case of deficiency of NADP^+^ for the decarboxylation reaction involving Gnd, excess 6PG is dumped through the Entner–Doudoroff pathway into lower glycolysis without the formation of pentoses. Inhibition of both dehydrogenases by NADPH represents a well-known paradox, since theoretically, increased expression of *zwf* and *gnd* genes in response to anabolic demand leads to increased NADPH generation and immediate inhibition of the enzymes [53].

Zwf activity is also determined by the energy status of the cell. It has been previously shown that Zwf is inhibited by ATP [54,55]. On the other hand, the metabolic impulse caused by the addition of glucose leads to a drop in ATP levels and a redirection of the carbon flow towards PPP [56].

### 3.2. Regulation of the Non-Oxidative Branch

The cascade of reactions of the non-oxidative branch of PPP can be carried out in two directions, both towards glycolysis and in the reverse anabolic direction towards the synthesis of pentose phosphates and has features of the enzyme composition for different groups of microorganisms (Figure 2). The main enzyme of the non-oxidative branch is transketolase, which catalyzes the reversible transfer of a ketol group between several donor and acceptor substrates [57]. Different bacterial species contain one to three transketolase isoforms, each of which functions under different physiological conditions [58,59,60,61]. In *E. coli* cells, as the most studied microorganism, there are two transketolase isoforms TktA and TktB. TktA is the major transketolase activity in the logarithmic growth phase of the culture, whereas TktB is active in the stationary growth phase [58]. The expression of *tktB* is under the positive control of the sigma factor RpoS, which acts during the transition from exponential growth in a rich medium to the stationary phase, accompanied by the depletion of complex nutrients. RpoS is also called “the master regulator of the general stress response in *Escherichia coli* “ [62]. The expression of the transketolase genes *tktB* and *tktC* is regulated in a similar manner in *Salmonella enterica* serovar Typhimurium [59].

Expression of the *tktA* gene increases under conditions of limited aeration [63], while FNR-dependent inactivation of oxy-branch gene expression synchronizes the switch of pentose synthesis through the reverse of the non-oxidative branch of PPP during anaerobiosis. At the moment, other mechanisms regulating *tktA* gene expression in *E. coli* are not known. The activity of the Tkt enzyme itself is regulated by various mechanisms including allosteric regulation by substrates and cofactors, metabolite levels, and redox modifications [64,65]. Transketolase is a thiamine pyrophosphate (TPP)-dependent enzyme. The active form of transketolase is a homodimer, each active site of which is occupied by one TPP molecule and a divalent cation such as Mg^2+^, Ca^2+^ or Mn^2+^ [66]. According to recent studies, oxidation (formation of sulfenic acid) of the Cys157 residue plays a special role in the activation of the Tkt homodimer [65]. The presence of Cys157 in many bacteria, yeasts, protozoa and plants, but not in animals such as *Homo sapiens* or *Mus musculus*, suggests the universality of this activation mechanism. Earlier studies demonstrated that double oxidation of Cys157 to sulfinic acid leads to inactivation of the enzyme [67]. In eukaryotic cells, regulation of Tkt activity is carried out by Akt1-dependent (protein kinase B-dependent) phosphorylation of the Thr382 residue. It was shown that amino acid starvation leads to a decrease in Akt1 and Tkt activity, limitation of the metabolic flux through the non-oxidative branch and the rate of purine synthesis [68].

Unlike the oxidative branch, the non-oxidative branch of PPP has several ways of converting F6P and GAP into pentoses (Figure 2). In the canonical pathway, the conversion of F6P and E4P is catalyzed by transaldolase Tal with the formation of S7P. Phylogenetically, transaldolases can be divided into three groups: classical transaldolases, which are found in archaea, bacteria, including cyanobacteria, and eukaryotes; MipB/TalC superfamily of truncated transaldolases, found in bacteria and archaea; and the third superfamily, transaldolases of higher plants, which have minimal sequence homology with the first two superfamilies, indicating a possible independent evolutionary origin [69]. In most microorganisms, Tal is represented by two isoforms. In *E. coli* cells, TalA transaldolase is part of a single operon with TktB under the control of the RpoS transcription factor [70]. Both enzymes function in the stationary growth phase or under metabolic stress. The main transaldolase activity is due to the TalB isoform. Interestingly, in *E. coli* the major forms of transketolase and transaldolase (*tktA* and *talB* genes, respectively) are monocistronic [71]. Transcriptional and post-transcriptional regulation of TalB in bacteria remain poorly understood. There is evidence that Tal activity is regulated by post-translational modifications such as phosphorylation. Two serine residues (37 and 226) have been identified as phosphorylation sites for *E. coli* TalB [72]. One of them (serine-226) is located in the active site [72] at the putative binding site of the phosphate group of substrates [73]. Therefore, phosphorylation at this site may alter the enzyme activity.

The alternative non-oxidative branch of the PPP implies the formation of S1,7bisP from E4P and dihydroxyacetone phosphate with the participation of Fba aldolase, followed by dephosphorylation by GlpX phosphatase to S7P. Phosphofructokinase Pfk works in the opposite direction, phosphorylating S7P in the first position. This type of non-oxidative branch is found in photosynthetics and methylotrophs, as mentioned above (Figure 2b,c). Traditionally, Fba aldolase, GlpX phosphatase and Pfk phosphofructokinase are considered glycolytic/gluconeogenic enzymes. Two classes of aldolases (FBAs) with different catalytic mechanisms have been described according to their amino acid sequences and designated as class I and class II FBAs, respectively [74]. Class II FBAs are commonly found in bacteria, archaea, and lower eukaryotes, including fungi and some green algae. Some bacterial species, including *E. coli*, have been reported to express both types of the enzyme. Class I FBAs are commonly found in higher eukaryotic organisms (animals and plants). Transcription of both aldolase genes as well as phosphofructokinase Pfk is negatively regulated by the catabolite repressor Cra [75]. On the other hand, the expression of aldolase and GlpX phosphatase genes activated by the cAMP receptor protein CRP. CRP directly transcriptionally regulates the expression of the (p) ppGpp synthetase/hydrolase RelA and SpoT and stimulates ubiquitous accumulation of (p) ppGpp under glucose-limiting conditions [76]. Thus, an alternative non-oxidative branch PPP requiring aldolase and phosphatase activity and resulting in formation of the intermediate S1,7bisP is also active under conditions of growth on gluconeogenic carbon sources.

The mutual conversion of pentose phosphates in the non-oxidative branch is carried out by R5P isomerase Rpi and Ru5P epimerase Rpe (Figure 1). R5P isomerase in *E. coli* cells is represented by two isoforms RpiA and RpiB, which differ greatly in the primary amino acid sequence and have different substrate specificities [77]. Interestingly, the expression of *rpiA* is under the negative control of the Lrp factor, which controls the genes of amino acid biosynthesis and catabolism, nutrient transport and other cellular functions, including one-carbon metabolism, while Lrp positively regulates the expression of *rpiB* [78,79]. This feature of the regulation of the expression of two Rpi isoforms indicates the difference in their physiological functions. It is known that RpiA works predominantly towards the formation of R5P from Ru5P, whereas RpiB carries out the reverse reaction [77,80].

The Ru5P epimerase gene *rpe* is located in a common operon with the shikimate pathway genes and is negatively regulated by the transcription factors ArgR and GlaR. ArgR controls arginine metabolism and transport in many bacteria and modulates *rpoS* expression [81]. GlaR (CsiR) regulates glutarate catabolism and activated during carbon starvation [82]. In addition, GlaR activates the expression of the second isoform of R5P isomerase RpiB. It should be noted that the activity of Rpe epimerase is critically dependent on divalent Fe^2+^ cations, which makes this enzyme vulnerable to oxidative stress [83].

Thus, the regulation of gene expression and enzyme activity of the oxidative and non-oxidative branches of the PPP involves two ways of synthesizing R5P—catabolic from glucose-6-phosphate and anabolic from fructose-6-phosphate and glyceraldehyde-3-phosphate, the choice between which is determined by the quantity and quality of carbon sources (glycolytic or gluconeogenic) and other nutritional elements, as well as environmental conditions such as aeration or stress factors of various nature. A feature of the catabolic pathway of pentose synthesis is the possibility of coupling ribose synthesis with the generation of NADPH.

## 4. Genetic Modification of PPP and Consequences for the Cell

PPP is one of the most studied parts of the metabolic network. PPP enzymes are considered as potential targets in the therapy of oncological diseases, infections, as well as in metabolic engineering in the creation of producer strains [84,85]. Deficiency of enzymes of the oxidative branch makes cells extremely vulnerable to oxidative stress [86,87]. In addition, inactivation of the *zwf* and *gnd* genes leads to a change in the metabolic flux and reverse of the non-oxidative branch of PPP, which becomes the main source of pentose phosphate synthesis [88]. In this case, the points of generation of NADPH in such mutants become isocitrate dehydrogenase and/or malate dehydrogenase depending on the carbon source [88]. However, the growth characteristics of these mutants indicate that the activity of the oxidative branch is not essential during growth on either rich or minimal media [88,89].

Of particular interest are studies devoted to ribose biosynthesis by the anabolic route. Sedoheptulose-1,7-bisphosphatase has been identified in yeast cells [90]; its activity provides a thermodynamically controlled pathway from trioses formed as a result of glycolysis to ribose synthesis. The flow through this pathway is regulated in response to the biosynthetic and redox requirements of the cell. This alternative pathway may explain the observed increase in S7P levels upon inhibition of the oxidative branch of PPP in eukaryotic cells [90]. On the other hand, it was found that transaldolase-deficient *E. coli* cells can grow on xylose as a carbon source through the activity of the glycolytic enzymes aldolase and phosphofructokinase to form the intermediate S1,7bisP [91]. Both processes are remarkably reminiscent of the non-oxidative branch of intestinal bacterial PPP (Figure 2d).

Follow up the general direction of evolution the non-oxidative branch of PPP allows research dedicated to the possibility of Gnd-mediated CO_2_ fixation in *E. coli* cells. It was noted that k_cat_ for Gnd with Ru5P as a substrate is two times higher than for ribulose-1,5-bisphosphate carboxylase/oxygenase (RuBisCo) in plants. It was shown that upon inactivation of *rpe* or *tkt*, as well as in a strain capable of metabolizing F6P only through the reversed non-oxidative branch, Ru5P can be acceptor of CO_2_ with the formation of 6-phosphogluconate metabolized through the Entner–Doudoroff pathway [92]. This autocatalytic pathway may have independent biotechnological applications.

The main function of the pentose phosphate pathway—the production of essential sugar phosphates, in particular R5P—can be replaced by the triple activity of only one universal enzyme, phosphoketolase PKT from *Bifidobacterium adolescentis* (Figure 3) [93]. Overexpression of PKT in *E. coli* cells with completely inactivated PPP via the *zwf* and *tktAB* genes deletions leads to prototrophic growth, catalyzing three sequential mechanisms: the distribution of Xu5P, F6P and, in particular, S7P. The presented studies are definitely interesting from the point of view of evolutionary biology and biochemistry and can find application in biotechnology. However, PPP in its current form with a given set of enzymes is optimal in an evolutionary sense, providing flexibility to the process of R5P synthesis in accordance with the anabolic status of the cell.

In a series of our studies on the effect of genetic modifications of PPP, as well as the identification of biosynthetic pathways involved in the metabolization of R5P on the resistance of *E. coli* cells to antibiotics, we clearly demonstrate that mutations of the PPP genes inducing unidirectional ribose synthesis lead to a dramatic increase in the sensitivity of bacteria to antibacterial drugs of various mechanisms of action associated with the accumulation of R5P (Figure 4) [13,14]. In *E. coli* cells, inactivation of the genes of the oxidative (*zwf*) and non-oxidative (*talAB*) branches of PPP leads to the reconstruction of an alternative anabolic synthesis of pentose phosphates with the participation of glycolytic enzymes aldolase Fba, phosphatase GlpX and phosphofructokinase Pfk (Figure 2b,c). The similar configuration of the non-oxidative branch of PPP was described in the work of Nakahigashi et al. during the growth of cells with a transaldolase activity deficiency on xylose as a carbon source [91]. It should be emphasized that the unidirectionality of such a metabolic pathway is imparted by the reaction of phosphorylation of S7P by phosphofructokinase Pfk, which requires the expenditure of ATP and complicates the return of excess pentoses to glycolysis. A feature of the anabolic synthesis of R5P is the uncoupling of this process from the generation of NADPH. The cell loses the ability to adjust the ribose pool in accordance with its anabolic potential expressed in reduced NADPH. An evolutionary feature of organisms with such a unidirectional way of synthesis of R5P is that its derivatives Ru5P or ribulose-1,5-bisphosphate act as acceptors of C1-carbon molecules (Figure 2b,c). The final result of such biosynthesis is the creation of nutritional elements in the form of 6-carbon sugars that can be metabolized in the traditional way through glycolysis. The action of antibacterial drugs causes inhibition of the main synthetic processes (synthesis of DNA, RNA, proteins), which leads to a dramatic increase in the level of R5P in triple mutants *Δzwf ΔtalAB* and increased bactericidal action compared to wild-type cells. This type of metabolism leads to an increase in the percentage of dead cells in the population during exponential growth. Activation of metabolic pathways such as the synthesis of purines or cell wall LPS components associated with ribose consumption leads to a decrease in the intracellular pool of R5P and restoration of tolerance to antibiotics (Figure 4) [14].

An unexpected result of this study was the discovery a latent source of R5P in *Δzwf ΔtalAB* cells, which turned out to be the process of degradation of purine nucleosides. Limitation of the final step of conversion of ribose-1-phosphate (R1P) to R5P by deletion of the phosphopentomutase gene *deoB* resulted in decreased accumulation of R5P and restoration of tolerance [14].

Inactivation of the R5P isomerase genes *rpiA* and *rpiB* also induces the reversal of the non-oxidative branch of PPP, accumulation of R5P and increased sensitivity to antibiotics [13]. The similar nature of sensitivity in the *Δzwf ΔtalAB* and *ΔrpiAB* mutants is evidenced by the same suppression mechanisms: increased metabolization of R5P and limitation of degradation of purine nucleosides as an additional source of ribose (Figure 4). In addition, excess ribose in the LB growth medium turned out to be toxic for the *ΔrpiAB* mutant. The obtained results allow us to conclude that the intensity of ribose synthesis should be adequate to the metabolic requests of the cell, otherwise excess R5P becomes lethal.

## 5. PPP as an Anabolic Sensor

The necessity to maintain cellular homeostasis in constantly changing environmental conditions implies the presence of a system of continuous detection in real time of such parameters as carbon sources, redox balance, aeration, stress factors. We assume that PPP has become such a system, which has undergone a long evolutionary path. The exclusive location of PPP in the metabolic network suggests the possibility of implementing an immediate choice: how and for what this or that nutrient will be spent. The main metabolic function of PPP is primarily the synthesis of pentose phosphates, especially R5P, the need for which is determined by the rate of biomass growth. With the emergence of aerobiosis and the possibility of rapid energy generation in the form of ATP in the process of oxidative phosphorylation, there was a need for continuous maintenance of the redox balance of the cell, where NADPH became the main source of reductive energy. Thus, direct coordination between the pools of NADPH and R5P reducing agents arose, carried out in the oxidative branch of PPP. Numerous studies show that, depending on physiological conditions, R5P synthesis can be either associated with NADPH generation or carried out independently, i.e., switching occurs between the two branches of PPP [3]. The most studied is the regulation of the activity of the oxidative branch under oxidative stress [4,8,94,95,96]. Considering the regulation of the main enzymes Zwf and Gnd, the activity of the oxidative branch increases in the presence of high levels of oxidized NADP^+^ [95]. During oxidative stress, the enzymes of the lower glycolysis glyceraldehyde-3-phosphate dehydrogenase (GAPDH) and pyruvate kinase (PK) are inhibited [97], at the same time cyclization of the metabolic flux through PPP allows the cell to generate large amounts of NADPH, with R5P becoming an intermediate metabolite. Such reorganization of PPP is observed both in bacterial cells during the response to oxidative stress [85,86,96,98], and in eukaryotic cells such as neutrophils during the oxidative burst, where NADPH is consumed for the production of superoxide [99] and erythrocytes, in which a pool of reduced glutathione is maintained for ROS quenching due to NADPH [98]. On the other hand, it was shown that during tumor progression, the main source of R5P becomes the reversed non-oxidative branch of PPP [2], while the deficiency of Zwf activity (less than 1%) does not reduce cancer incidence [100,101]. It is noteworthy that in actively proliferating cells, along with PPP, a comparable contribution to the generation of NADPH is made by serine-controlled one-carbon metabolism, in which the oxidation of methylenetetrahydrofolate to 10-formyltetrahydrofolate is associated with the reduction of NADP^+^ to NADPH (Figure 5) [102].

Thus, folate metabolism for proliferating cells acquires an additional context. In our studies on *E. coli* cells, we have shown that inactivation of the *zwf* gene leads to the generation of NADPH via the serine–glycine pathway coupled with purine biosynthesis [14]. Gene serine hydroxymethyltransferase *glyA* belongs to the purine regulon under the control of the repressor PurR. Apparently, the reversed non-oxidative branch under conditions of intense cell proliferation becomes the main source of R5P, while the serine–glycine pathway coupled with purine nucleoside biosynthesis and active ribose metabolization becomes the preferred way of generating NADPH to maintain anabolic processes. A similar situation can be observed in photosynthetics during photorespiration, when unused ribose-1,5-bisphosphate is oxidized to form 2-phosphoglycolate, which must be immediately metabolized via the serine–glycine pathway [103]. An interesting fact confirming that anabolic synthesis of ribose is preferable under conditions of active biomass growth is the correlation of the intensity of the flux through the reversed non-oxidative branch with the expression levels of ribosomal proteins in yeast cells [90].

To summarize the above, the main functional feature of PPP is the ability to switch between two pathways for the synthesis of pentose phosphates, and if the need for ribose and NADPH is high enough, a simultaneous flow through the oxidative and non-oxidative branches towards the synthesis of pentoses is not excluded.

## 6. Conclusions

R5P is a critical metabolite for any living cell. The process of pentose phosphate synthesis has undergone a long evolution and has taken shape in the PPP, which allows solving several metabolic problems at once. The classical concept of the canonical pentose phosphate pathway implies the coupling of R5P biosynthesis processes with the generation of NADPH in the oxidative branch and the possibility of returning the excess of all intermediates to glycolysis through the non-oxidative branch [104]. Regulation of the metabolic flow through the oxidative branch is carried out at all levels, from gene transcription to post-translational interactions, which allows the cell to assess the quantity and quality of nutrients and other environmental parameters in real time, optimizing anabolic processes (Figure 6).

Among all the regulatory mechanisms, the NADP^+^ pool has particular importance for the activity of the enzymes of the oxidative branch Zwf and Gnd [52,105]. The NADP^+^/NADPH ratio marks the intensity of constructive metabolism, essentially the rate of growth and division, which allows for the precise determination of the required amount of R5P. Under oxidative stress, the cell can generate a sufficient amount of reducing agents without creating excess pentose phosphates. However, a change in physiological conditions implies a switch between two branches of PPP, i.e., two pathways for the synthesis of R5P can be distinguished: catabolic with the participation of oxyPPP enzymes and anabolic, implying the reverse of the non-oxidative branch of PPP. Apparently, anabolic synthesis of R5P, being primary from the point of view of biochemical evolution, is preferable during intensive biomass growth, when the need for ribose exceeds the need for NADPH [1,2]. R5P synthesized anabolic must be immediately metabolized in the processes of biosynthesis of RNA, DNA and other derivatives. Ribose acts as a factor in the consolidation of anabolic processes associated with cell proliferation and biomass growth. Our results clearly demonstrate that under conditions of R5P consumption limitation caused by the action of antibacterial drugs, a bacterial cell with a similar type of pentose phosphate metabolism becomes supersensitive to these drugs. A direct correlation is observed between the intracellular R5P pool and antibiotic sensitivity. Intensive unregulated synthesis underlies the indirect toxic effect of excess R5P on the bacterial cell. We believe that this effect of ribose can become the basis for new antibiotic agents.

## Figures and Tables

**Figure 1 cells-14-01775-f001:**
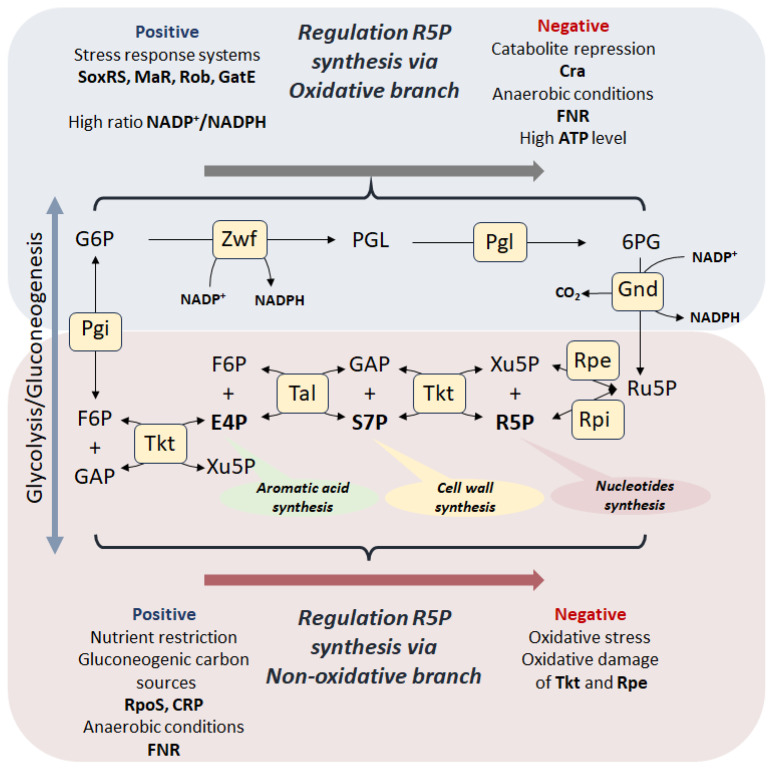
Synthesis of R5P in canonical PPP. The canonical PPP consists of an oxidative branch (highlighted in blue) in which the key reactions of ribulose-5-phosphate (Ru5P) synthesis, accompanied by the reduction of NADP^+^, are catalyzed by the enzymes glucose-6-phosphate dehydrogenase (Zwf) and phosphogluconate dehydrogenase (Gnd). 6-phosphogluconolactone (PGL) conversion to 6-phosphogluconate (6PG) occurs both spontaneously and with the participation of 6-phosphogluconolactonases Pgl. The oxidative branch of the PPP is considered irreversible due to the loss of CO_2_ during the conversion of 6-phosphogluconate (6PG) to ribulose-5-phosphate (Ru5P). The non-oxidative branch (highlighted in pink) begins with the conversion of ribulose-5-phosphate to ribose-5-phosphate (R5P) by ribose-5-phosphate isomerase (Rpi) or to xylose-5-phosphate (Xu5P) by ribulose-5-phosphate epimerase (Rpe). R5P and Xu5P can be recycled back into glycolysis as fructose-6-phosphate (F6P) and glyceraldehyde-3-phosphate (GAP) via a reversible reaction cascade involving transketalase (Tkt) and transaldolase (Tal). R5P can be synthesized via oxidative branch, which is positively regulated on transcriptional level by the SoxRS, Mar, and Rob stress response systems and high ratio NADP^+^/NADPH, and via the reversed non-oxidative branch under conditions of metabolic transition caused by depletion of glycolytic carbon sources and limitation of aeration.

**Figure 3 cells-14-01775-f003:**
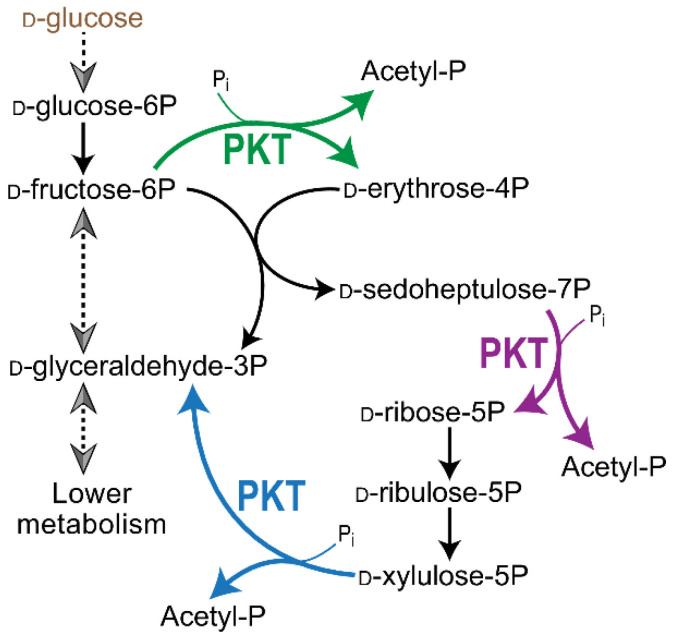
The activity of PKT phosphoketolase from *Bifidobacterium adolescentis* completely replaces the enzymes of the non-oxidative branch of PPP, carrying out the synthesis of R5P [93].

**Figure 4 cells-14-01775-f004:**
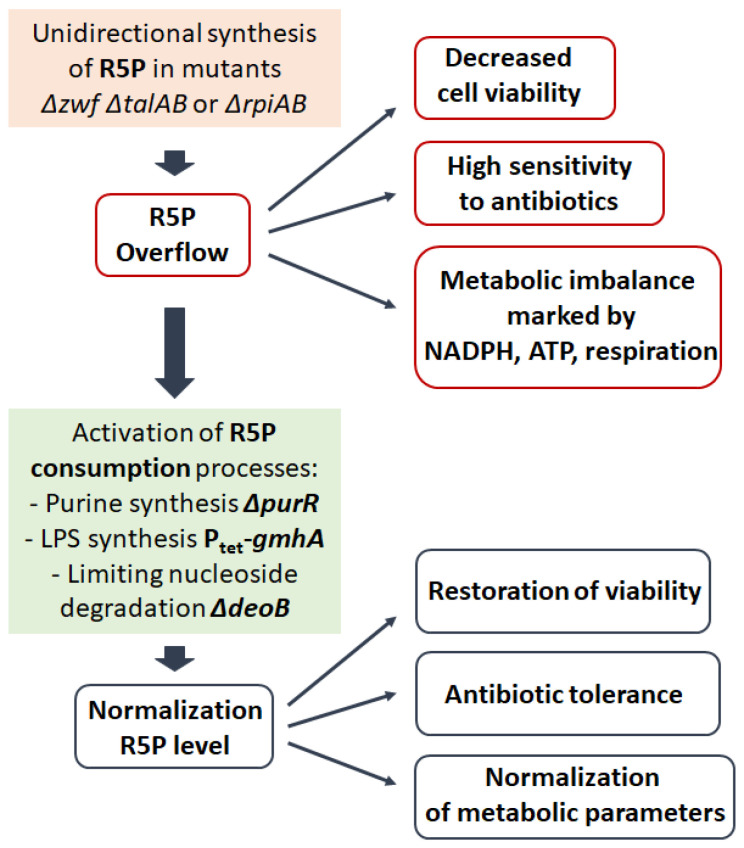
Synthesis of R5P underlies metabolic stability and viability of bacterial cells. Unidirectional synthesis of R5P through the non-oxidative branch of the PPP induced by mutations in the genes *zwf, talAB* or *rpiAB* reduces cell viability and their resistance to antibiotics. *purR*—encodes a purine regulon repressor. *gmhA*—encodes S7P isomerase, the enzyme of the first stage of ADP-heptose synthesis of LPS. *deoB*—encodes phosphopentomutase that converts R1P to R5P during nucleoside degradation.

**Figure 5 cells-14-01775-f005:**
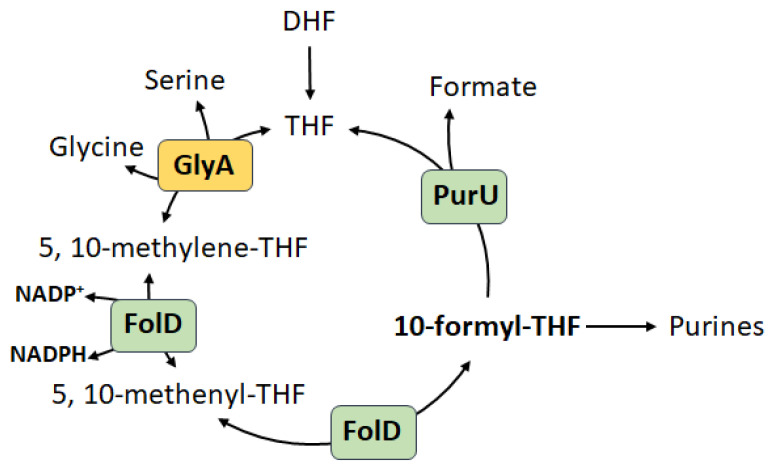
Folate-dependent NADPH production in *E. coli* cells. Against the background of inactivation of the oxidative branch of the PPP mediated by the *zwf* deletion, the folate cycle, coupled with the biosynthesis of purines, becomes the main source of NADPH. DHF—dihydrofolate; THF—tetrahydrofolate; GlyA—serine hydroxymethyltransferase; FolD—bifunctional methylenetetrahydrofolate dehydrogenase/methenyltetrahydrofolate cyclohydrolase; PurU—formyltetrahydrofolate deformylase.

**Figure 6 cells-14-01775-f006:**
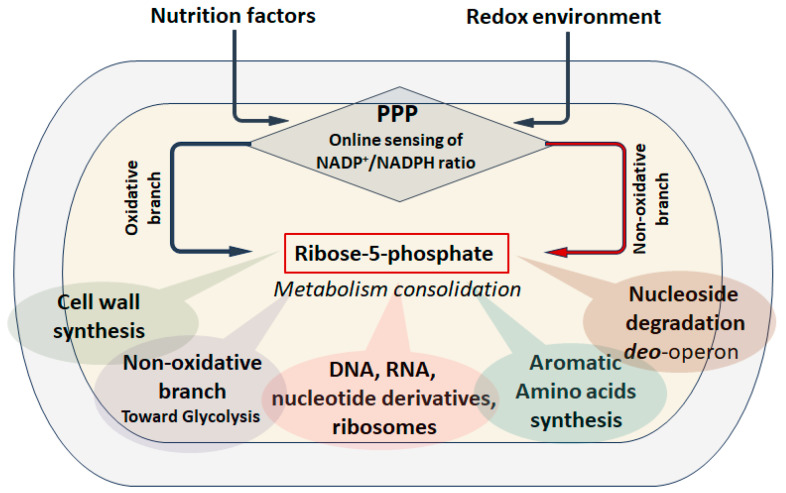
PPP as an anabolic sensor. PPP allows monitoring in real time the quantity and quality of carbon sources and the redox environment, continuously detecting the ratio NADP^+^/NADPH, which allows choosing the optimal pathway for synthesizing R5P through the oxidative or non-oxidative branch. The R5P is a factor which consolidate of the metabolic network involved in growth program and its pool must strictly correspond to the anabolic status of the cell.

## Data Availability

This study did not report any data.

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
