# Peer review of "Biosynthesis of Ribose-5-Phosphate—Metabolic Regulator of Escherichia coli Viability"

_cells, 2025, doi:10.3390/cells14221775_

Round 1

Reviewer 1 Report

Comments and Suggestions for Authors

In their paper entitled « Biosynthesis of ribose-5-phosphate – metabolic regulator of

Escherichia coli viability » the authors report the diversity of the organization, regulation and

evolutionary origin of the oxidative/catabolic part and the non-oxidative/anabolic part of the

Pentose Phosphate Pathway (PPP) in various organisms and in different nutritional and

environemental conditions (anaerobiosis, stress, carbon source etc...). The oxidative part of the

PPP is irreversible and generates NADPH, an important co-factor protecting the cells against

oxidative stress. In contrast, the non-oxidative part of the PPP is reversible and can be used to fix

CO2. The authors also mentioned a puzzling link between high ribose 5 P levels and enhanced

sensitivity to antibiotics that are known to generate oxidative stress. This enhanced sensitivity

that does not seem to be linked to low NADPH availability but rather to high R5P accumulation

that results from the arrest of anabolic processes (DNA, RNA or proteins synthesis), a

consequence of antibiotics action. However the molecular targets of R5P remain to be defined

to support the hypothesis of R5P toxicity. Furthermore, it would be nice to know what is the

impact of the presence of antibiotics on the relative expression of the two branches of the PPP.

Otherwise, the manuscript is clear, well structured and well written.

I thus have only minor comments

line 16 : I will suggest to cancel « reduced » in front of NADPH

line 354 : I will suggest to use NADH rather than « equivalents »

line 385:providing flexibility to the process of R5P synthesis in accordance with the metabolic

status of the cell.

Line 461 « … the lower part of glycolysis »

Line 468 « … thanks to NADPH equivalents. »

Figure 5 is unclear since the authors show a pathway present in E. coli but in the legend they

comments on a system present in eukaryotic cells that is not shown in the figure (Fdh). This

eukaryotic route should perhaps be added in the figure using dotted lines.

Figure 6 Would not it be better to replace « Non-oxidative branch / glycolytic direction » by

« Toward Glycolysis »

Author Response

Reviewer #1:

In their paper entitled « Biosynthesis of ribose-5-phosphate – metabolic regulator of

Escherichia coli viability » the authors report the diversity of the organization, regulation and evolutionary origin of the oxidative/catabolic part and the non-oxidative/anabolic part of the Pentose Phosphate Pathway (PPP) in various organisms and in different nutritional and environemental conditions (anaerobiosis, stress, carbon source etc...). The oxidative part of the PPP is irreversible and generates NADPH, an important co-factor protecting the cells against oxidative stress. In contrast, the non-oxidative part of the PPP is reversible and can be used to fix CO2. The authors also mentioned a puzzling link between high ribose 5 P levels and enhanced sensitivity to antibiotics that are known to generate oxidative stress. This enhanced sensitivity that does not seem to be linked to low NADPH availability but rather to high R5P accumulation that results from the arrest of anabolic processes (DNA, RNA or proteins synthesis), a consequence of antibiotics action. However the molecular targets of R5P remain to be defined to support the hypothesis of R5P toxicity. Furthermore, it would be nice to know what is the impact of the presence of antibiotics on the relative expression of the two branches of the PPP. Otherwise, the manuscript is clear, well structured and well written.

We sincerely appreciate the reviewer’s thoughtful and constructive comments. All the issues raised have been thoroughly addressed as follows:

«Furthermore, it would be nice to know what is the impact of the presence of antibiotics on the relative expression of the two branches of the PPP».

Studying the specifics of PPP regulation and the processes associated with R5P metabolization is our current area of ​​research interest. We hope to obtain new data, including on the role of antibiotics in altering the metabolic flux in the PPP.

I thus have only minor comments

line 16 : I will suggest to cancel « reduced » in front of NADPH

As recommended by the reviewer, the text has been corrected

line 354 : I will suggest to use NADH rather than « equivalents »

Corrected

line 385: providing flexibility to the process of R5P synthesis in accordance with the metabolic status of the cell.

The text has been corrected

Line 461 « … the lower part of glycolysis »

lower glycolysis

The text has been corrected

Line 468 « … thanks to NADPH equivalents. »

 Corrected

Figure 5 is unclear since the authors show a pathway present in E. coli but in the legend they

comments on a system present in eukaryotic cells that is not shown in the figure (Fdh). This

eukaryotic route should perhaps be added in the figure using dotted lines.

Figure 5 has been adjusted as per reviewer's recommendation.

Figure 6 Would not it be better to replace « Non-oxidative branch / glycolytic direction » by

« Toward Glycolysis »

As suggested by the reviewer, Figure 6 has been сcorrected

Reviewer 2 Report

Comments and Suggestions for Authors

The manuscript is a review of the formation of ribose-5-phosphate.  I find the work disorganized, and I am not sure what the message is.  There are two general problems with the work. 

One is that the authors use English terms that are not quite correct.   For example, the term “variant” (used often….Lines 111, 143, 282, 283, 301, 304, etc.) is not typically applied to ‘variant pathways’.  And, the authors use the term to describe alternatives to those “variants” listed.  In a sense, this makes everything a variant.  Perhaps the word “alternative” would be better.  But, there are many other examples in the manuscript in which a reader would be confused or mislead by the word or phrase used.  Some are found in the “Other comments”.

Another general problem is that some of the work seems unnecessarily speculative. For example, although the focus of the work is the pathways associated with ribose-5-phosphate formation, there is a lot of speculative text regarding the process of evolution.  For example, the authors speculate that “the emergence of the oxidative branch of PPP was associated with the appearance of molecular oxygen as a by-product of photosynthesis and the need to protest the enzymatic apparatus of the cell from osidatve damage.”  (line 152-154).  This sentence is not supported by a specific reference, and seems beyond the scope of a review on the biochemical formation of ribose-5-phosphate.  (Similar statements in lines 173-174).  The presence of discussion surrounding evolution seems out of scope for the purpose of the manuscript.

Other comments:

Line 53, 71, 220, 470, 486, 494:  use “reversible non-oxidative…” instead of “reverse non-oxidative”

Line 101:  change “A typical example are” to “Typical examples are”

Line 132-134:  The authors reference figure 2c for the statement “…converting fructose-6-phosphate to Ru5P and formaldehyde.”  Figure 2c does not show the conversion of fructose-6-phosphate to Ru5P and formaldehyde.

Figure 2 would benefit from drawing in such a way as to make all the ‘starting substrates’ appear together.  For example, in Figure 2b the product is 3 x Ru5P.  The substrates are F6P, DHAP and two GAP, but one has to hunt the figure to find them.  It is not made clear why a portion of each figure appears in a ‘box’, and why some of the enzyme steps are written in a blue box.  Finally, some of the curves are completed rather imprecisely (e.g., the arrow between Xy5P + R5P and 3 x Ru5P is sloppily transected by the curve from Xu5P).

Line 187:  “conserved” instead of “conservative”?

Line 211:  “gluconeogenic enzymes” instead of “gluconeogenic enzyme”.  What specific enzymes are activated?

Line 221, 257: “transketolase” instead of “transketalase”

Line 226-228:  combine sentences to, for example, “The intermediates of PPP including 6PG, X5P, R5P, Ru5P, E4P and S7P have been shown to inhibit the glycolytic enzyme PGI [41-44]”.  Note that the phrase “intermediate product” is an oxymoron.

Line 233-234:  It is not clear what is meant by “competition between the enzymes Zwf and Gnd”.  True, both enzymes use NADP as a substrate.  However, the enzymes do not appear at a branchpoint, but rather are sequential in the oxidative branch of the PPP.  Thus, it is misleading to state that there is “competition” between them.  Their action instead is sequential, with one occurring first (Zwf) before the other can occur.  The fact that they use the same substrate merely indicates which enzyme would control the overall rate of the linear pathway, and does not indicate that the two enzymes are in competition.

Line 239-241:  It is not unusual for the product of an enzymatic reaction to inhibit that enzyme, modulating the flux through that enzyme until the product is utilized elsewhere in metabolism.  Why is this feature of the PPP of interest?

Line 244-245:  It is not clear what is meant by “metabolic impulse”.  This sentence does not contain a reference.

Line 296: change “remains” to “remain”

Line 449:  Does the rate of biomass growth determine the rate of R5P synthesis, or does the rate of R5P determine the rate of biomass growth?

Line 464:  R5P is always a metabolite, and since it likely does not accumulate, at least not to the extent of becoming an extracellular product, R5P could likely always be thought of as an “intermediate metabolite”.  If it is always an intermediate metabolite, then it is not possible for it to “become” an intermediate metabolite. 

Section 4.  There have been quite a lot of studies knocking out genes associated with the PPP, and the authors only cite a very few of them.  It is not clear what the lessons are.

Figure 5.  Very confusing as to what this figure shows.  The first sentence of the caption states what occurs when a zwf deletion occurs in E. coli cells, but does not indicate what the figure shows.  The second sentence states what occurs in eukaryotes and that CO2 is lost.  No CO2 appears in the figure. What does this figure show?  Is it for E. coli or for eukaryotes?  What is the purpose of the figure in the context of the focus of the review, namely R5P formation?  Is it merely sufficient to state that in the absence of the oxidative branch of the PPP, organisms find other means to generate NADPH?

The choice provided was either "minor revision" or "major revision".  I selected "major revision", but a more accurate recommendation would be "moderate revision"

Comments on the Quality of English Language

As noted in review, the specific word choice is poor in several instances.  I suggest defining some terms carefully in the "Intro", and then using those terms throughout.  Also, have a native english-speaking with some subject matter knowledge review word choice.

Author Response

Reviewer #2:

We sincerely appreciate the reviewer thorough analysis of the manuscript. In response to the concerns and suggestions raised, manuscript correction has been conducted, as detailed below.

The manuscript is a review of the formation of ribose-5-phosphate.  I find the work disorganized, and I am not sure what the message is.  There are two general problems with the work. 

In this review, we examine the process of ribose 5-phosphate synthesis and its regulation in various groups of organisms within the context of their physiological characteristics, formed during biochemical evolution. The main idea we formulate in our work is that ribose 5-phosphate synthesis is a point of vulnerability for cells, and imbalance between ribose-5-phosphate synthesis and consumption initiates spontaneous cell death and reduces their stress resistance [DOI: 10.1128/mbio.00654-25].

One is that the authors use English terms that are not quite correct.   For example, the term “variant” (used often….Lines 111, 143, 282, 283, 301, 304, etc.) is not typically applied to ‘variant pathways’.  And, the authors use the term to describe alternatives to those “variants” listed.  In a sense, this makes everything a variant.  Perhaps the word “alternative” would be better.  But, there are many other examples in the manuscript in which a reader would be confused or mislead by the word or phrase used.  Some are found in the “Other comments”.

In accordance with reviewer's recommendations, the text has been corrected.

Another general problem is that some of the work seems unnecessarily speculative. For example, although the focus of the work is the pathways associated with ribose-5-phosphate formation, there is a lot of speculative text regarding the process of evolution.  For example, the authors speculate that “the emergence of the oxidative branch of PPP was associated with the appearance of molecular oxygen as a by-product of photosynthesis and the need to protest the enzymatic apparatus of the cell from oxidative damage.”  (line 152-154).  This sentence is not supported by a specific reference, and seems beyond the scope of a review on the biochemical formation of ribose-5-phosphate.  (Similar statements in lines 173-174).  The presence of discussion surrounding evolution seems out of scope for the purpose of the manuscript.

We agree with the reviewer's opinion that our statements are somewhat speculative. However, these assumptions are based on obvious differences in the nature of enzymatic reactions in two branches of the PPP and their regulation.  Modern PPP actually consists of two main elements: the oxidative branch, which is part of the SoxRS-Mar oxidative stress defense system, and the non-oxidative branch, which initially, in addition to the synthesis of R5P, was involved in the fixation of C1 compounds. Metagenomic analysis shows that the emergence of enzymes (catalases, superoxide dismutases, peroxidases etc. [https://doi.org/10.1016/j.tim.2020.10.001]) which protect cell from oxidative stress and, accordingly, their regulatory systems is associated with the onset of oxygenic photosynthesis in cyanobacteria), and characteristics of the PPP in the most ancient representatives of the prokaryotic world, as demonstrated by researchers whose work we cite [doi:10.1002/msb.20145228, https://doi.org/10.1111/brv.12140, DOI: 10.1038/s42255-023-00863-2].

Other comments:

Line 53, 71, 220, 470, 486, 494:  use “reversible non-oxidative…” instead of “reverse non-oxidative”

It is generally accepted that the non-oxidative branch serves to return excess pentose phosphates to glycolysis. However, we are considering a situation where the flow through it is directed toward pentose phosphate synthesis. We are not talking about reversibility, but rather about a reverse of the flow (unidirectional synthesis R5P). The term "reversed non-oxidative branch" implies an anabolic direction, as in the Calvin cycle.

Line 101:  change “A typical example are” to “Typical examples are”

Corrected

Line 132-134:  The authors reference figure 2c for the statement “…converting fructose-6-phosphate to Ru5P and formaldehyde.”  Figure 2c does not show the conversion of fructose-6-phosphate to Ru5P and formaldehyde.

Text has been corrected. In archaea, Ru5P is involved in the assimilation of formaldehyde.

Figure 2 would benefit from drawing in such a way as to make all the ‘starting substrates’ appear together.  For example, in Figure 2b the product is 3 x Ru5P.  The substrates are F6P, DHAP and two GAP, but one has to hunt the figure to find them.  It is not made clear why a portion of each figure appears in a ‘box’, and why some of the enzyme steps are written in a blue box.  Finally, some of the curves are completed rather imprecisely (e.g., the arrow between Xy5P + R5P and 3 x Ru5P is sloppily transected by the curve from Xu5P).

In Figure 2, we present the reactions and enzymes of the non-oxidative branch of the PPP for different groups of organisms: a) eukaryotes and aerobic and facultatively anaerobic bacteria, b) photosynthetic (cyanobacteria and plants), c) methylotrophs (archaea), d) anaerobic intestinal bacteria.  The aldolase-catalyzed stage of S1,7BP formation is indicated in a «box». The formation of S1,7BP is of fundamental importance for the reversibility of the non-oxidative branch. The reactions of the non-oxidative branch produce 3 pentoses (2 Xu5P, which can be converted to Ru5P by ribulose-5-phosphate epimerase, and one R5P). The arrows have been corrected.

Line 187:  “conserved” instead of “conservative”?

Corrected

Line 211:  “gluconeogenic enzymes” instead of “gluconeogenic enzyme”.  What specific enzymes are activated?

Sentence has been corrected: «Under conditions of glucose limitation and a decrease in the fructose-1-phosphate pool, “catabolite repressor activator” Cra activates the gene expression of gluconeogenic enzymes phosphoenolpyruvate synthetase Pps and fructose-1,6-bisphosphatase Fbp, Krebs cycle enzyme isocitrate dehydrogenase Icd  and the glyoxylate shunt enzymes isocitrate lyase AceA and malate synthase A AceB, and simultaneously inactivates the expression of zwf and genes of the Entner-Doudoroff pathway enzymes phosphogluconate dehydratase Edd and aldolase Eda, redirecting the carbon flow towards gluconeogenesis and, as a consequence, the synthesis of R5P is carried out through the anabolic pathway from fructose-6-phosphate and glyceraldehyde-3-phosphate by reversing the non-oxidative branch of PPP».

Line 221, 257: “transketolase” instead of “transketalase”

Corrected

Line 226-228:  combine sentences to, for example, “The intermediates of PPP including 6PG, X5P, R5P, Ru5P, E4P and S7P have been shown to inhibit the glycolytic enzyme PGI [41-44]”.  Note that the phrase “intermediate product” is an oxymoron.

Corrected

Line 233-234:  It is not clear what is meant by “competition between the enzymes Zwf and Gnd”.  True, both enzymes use NADP as a substrate.  However, the enzymes do not appear at a branchpoint, but rather are sequential in the oxidative branch of the PPP.  Thus, it is misleading to state that there is “competition” between them.  Their action instead is sequential, with one occurring first (Zwf) before the other can occur.  The fact that they use the same substrate merely indicates which enzyme would control the overall rate of the linear pathway, and does not indicate that the two enzymes are in competition.

The ability to synthesize R5P via the oxidative pathway is determined by the level of NADP+, i.e., the cell's anabolic potential. The oxidative branch has a branchpoint: 6-phosphogluconate, which can be metabolized via the Entner-Doudoroff pathway when NADP+ is deficient. It is not only a question of the overall rate of the linear pathway but also whether R5P will be synthesized at all, since the insufficient level of NADP+ for Gnd does not allow the decarboxylation reaction to occur.

Line 239-241:  It is not unusual for the product of an enzymatic reaction to inhibit that enzyme, modulating the flux through that enzyme until the product is utilized elsewhere in metabolism.  Why is this feature of the PPP of interest?

There are 4 points of generation of NADP+ in the cell and only one of them is associated with the synthesis of R5P. The regulation of oxidative branch enzymes by the principle of negative feedback is of interest in the context of the phenomenon of the direct or indirect toxic effect of excess R5P (the mechanism is not yet clear) that we discovered [doi: 10.1128/mbio.00654-25. Epub 2025 Jul 2].

Line 244-245:  It is not clear what is meant by “metabolic impulse”.  This sentence does not contain a reference.

An increase in exogenous glucose levels leads to a decrease in ATP levels, which is explained by a change in the mode of glycolysis; this metabolic change is known as the ATP paradox [https://doi.org/10.1074/jbc.M410479200]. Furthermore, under conditions of excess glucose, the flow from glycolysis is redirected to the oxidative branch of the PPP. Reference added.

Line 296: change “remains” to “remain”

Corrected

Line 449:  Does the rate of biomass growth determine the rate of R5P synthesis, or does the rate of R5P determine the rate of biomass growth?

Considering that excess R5P is toxic to the cell, the flow rate through the PPP (the amount of synthesized R5P) must correspond to the rate of biomass growth. We consider the PPP as a sensor that regulates the synthesis of R5P in accordance with the anabolic status expressed by the NADP+/NADPH ratio.

Line 464:  R5P is always a metabolite, and since it likely does not accumulate, at least not to the extent of becoming an extracellular product, R5P could likely always be thought of as an “intermediate metabolite”.  If it is always an intermediate metabolite, then it is not possible for it to “become” an intermediate metabolite. 

According to our data, R5P cannot be considered as exclusively an “intermediate metabolite”. There are modifications of PPP, when R5P is synthesized only in one direction without the possibility of returning to central metabolism. Reconstruction of such a unidirectional synthesis of R5P in the E. coli cell allowed us to discover the toxic effect of excess R5P in conditions where its rapid metabolization is impossible.

Section 4.  There have been quite a lot of studies knocking out genes associated with the PPP, and the authors only cite a very few of them.  It is not clear what the lessons are.

We agree with the reviewer that only a portion of the studies devoted to knockouts were cited. We consider those modifications of the PPP that are important in the context of our work.

Figure 5.  Very confusing as to what this figure shows.  The first sentence of the caption states what occurs when a zwf deletion occurs in E. coli cells, but does not indicate what the figure shows.  The second sentence states what occurs in eukaryotes and that CO2 is lost.  No CO2 appears in the figure. What does this figure show?  Is it for E. coli or for eukaryotes?  What is the purpose of the figure in the context of the focus of the review, namely R5P formation?  Is it merely sufficient to state that in the absence of the oxidative branch of the PPP, organisms find other means to generate NADPH?

The figure 5 shows folate-dependent NADPH production. We discovered that this pathway of NADPH generation, coupled with purine synthesis, becomes the primary pathway upon inactivation of the oxidative branch of the PPP. This is important in the context of a new type of metabolism resulting from the anabolic synthesis of pentose phosphates. Figure 5 has been adjusted as per reviewer's recommendation.

The choice provided was either "minor revision" or "major revision".  I selected "major revision", but a more accurate recommendation would be "moderate revision"

Comments on the Quality of English Language

As noted in review, the specific word choice is poor in several instances.  I suggest defining some terms carefully in the "Intro", and then using those terms throughout.  Also, have a native english-speaking with some subject matter knowledge review word choice.

Reviewer 3 Report

Comments and Suggestions for Authors

This paper introduces and summarize the R5P biosynthesis pathways among different domains of microorganism. Overall, more advanced strategies and metabolic regulator efficiencies should be well summarized, introduced and discussed. The writing and quality need be further improved. Major revision is suggested as below:

1. In "Introduction", the Ribose 5-phosphate function and application need be well introduced. More details should be provided.

2. The key enzymes and genes should be introduced 

3. The effects or carbon source and additives should be introduced and discussed in detail.

4. The information and data about enhancing the enzymatic activity, the product yield and the substrate titer need be introduced and summarized.

5. The prospect of the future work need be well summarized and described. The machine learning, AI-assisting and other new technologies should be considered.  More advanced strategies are required to provide.

Author Response

Reviewer #3:

This paper introduces and summarize the R5P biosynthesis pathways among different domains of microorganism. Overall, more advanced strategies and metabolic regulator efficiencies should be well summarized, introduced and discussed. The writing and quality need be further improved. Major revision is suggested as below:

We sincerely appreciate the reviewer’s thoughtful and constructive comments.

  1. In "Introduction", the Ribose 5-phosphate function and application need be well introduced. More details should be provided.

The text has been supplemented in accordance with the reviewer's wishes. See the lines 34-37 and 58-61.

  1. The key enzymes and genes should be introduced 

All enzymes and genes of the PPP are shown in Figure 1, which is cited in the introduction (line 37).

  1. The effects or carbon source and additives should be introduced and discussed in detail.

The aim of this review is to examine the genetic control of PPP rather than the relevant physiological characteristics of such strains, including carbon sources and additives.

  1. The information and data about enhancing the enzymatic activity, the product yield and the substrate titer need be introduced and summarized.

The aim of this work is not to describe various approaches for constructing producers and to consider their physiological characteristics.

  1. The prospect of the future work need be well summarized and described. The machine learning, AI-assisting and other new technologies should be considered.  More advanced strategies are required to provide.

Our research interests include studying the genetic control of bacterial central metabolism. The reviewer's proposed addition to our review is beyond our expertise and requires the participation of specialists from other fields, such as bioinformatics. Examples of such reviews include the following publications: DOI: 10.1038/s44259-025-00085-4, https://doi.org/10.1016/j.jpha.2025.101437, doi: 10.3389/fmicb.2023.1254073, https://doi.org/10.1111/1751-7915.70131. In this review, we examine the genetic regulation of PPP and ribose biosynthesis in the context of our recent data on the role of ribose in bacterial cell viability and stress resistance. Shifting the focus may distract and mislead the reader.

Round 2

Reviewer 2 Report

Comments and Suggestions for Authors

The work is improved and appropriate clarifications made.  I encourage the authors to consider increasing the number of citations, even to the extent of being a little redundant in citing works  to make the reference clear.  One value of a review is its comprehensive citations.  Some other suggestions to improve the work:

Line 179:  delete comma after “with”

Line 230:  phrase “not only and not so much” is confusing.  I am expecting a subsequent phrase “but also”.  That is, if a phenomenon depends “not only” on one factor, then it should also depend on another factor.  In this sentence that other factor is not clearly stated.  Would it be sufficient to delete this phrase? 

Line 264:  “transketolase” instead of “transketalase”

Line 360:  Suggest stating “NADPH” instead of “equivalents”

Line 384, Figure 3 legend:  “phosphoketolase” instead of “phosphoketalase”

Line 385:  tktA and/or tktB genes?

Line 434:  This paragraph does not have a citation.  Should include a citation so there is no ambiguity as to what ‘study’ is being described.

Author Response

The work is improved and appropriate clarifications made.  I encourage the authors to consider increasing the number of citations, even to the extent of being a little redundant in citing works to make the reference clear.  One value of a review is its comprehensive citations.  Some other suggestions to improve the work:

Several new references have been added as recommended by the reviewer: 24, 27, 30, 41, 42, 70, 84.

Line 179:  delete comma after “with”

Sentence has been corrected

Line 230:  phrase “not only and not so much” is confusing.  I am expecting a subsequent phrase “but also”.  That is, if a phenomenon depends “not only” on one factor, then it should also depend on another factor.  In this sentence that other factor is not clearly stated.  Would it be sufficient to delete this phrase? 

Sentence has been corrected: “The coordination of metabolic fluxes between glycolysis and the oxidative branch of PPP has enormous physiological importance. It should be noted, that the availability of the precursor for both pathways, glucose-6-phosphate, and the Michaelis-Menten constants for the enzymes Zwf (Km=174 µM) [41] and glucose-6-phosphate isomerase (Pgi) (Km=1018 µM) are not the only factors determining the direction of flow.”

Line 264:  “transketolase” instead of “transketalase”

Corrected

Line 360:  Suggest stating “NADPH” instead of “equivalents”

Corrected

Line 384, Figure 3 legend:  “phosphoketolase” instead of “phosphoketalase”

Corrected

Line 385:  tktA and/or tktB genes?

Sentence has been corrected. In the cited work, both transketolase genes tktA and tktB were inactivated, and those cells were completely devoid of transketolase activity.

Line 434:  This paragraph does not have a citation.  Should include a citation so there is no ambiguity as to what ‘study’ is being described.

The link has been added.

Reviewer 3 Report

Comments and Suggestions for Authors

This revised version can be accepted as it is.

Author Response

This revised version can be accepted as it is.

We thank the reviewer for carefully reading our work.